# An Emerging Innovative UV Disinfection Technology (Part II): Virucide Activity on SARS-CoV-2

**DOI:** 10.3390/ijerph18083873

**Published:** 2021-04-07

**Authors:** Gabriele Messina, Alessandro Della Camera, Pietro Ferraro, Davide Amodeo, Alessio Corazza, Nicola Nante, Gabriele Cevenini

**Affiliations:** 1Department of Molecular and Developmental Medicine, University of Siena, 53100 Siena, Italy; pietro.ferraro@unisi.it (P.F.); davide.amodeo@student.unisi.it (D.A.); nicola.nante@unisi.it (N.N.); 2Department of Molecular and Developmental Medicine, Post Graduate School of Public Health, University of Siena, 53100 Siena, Italy; alessa.dellacamera@student.unisi.it; 3SAES Getters S.p.A., Viale Italia 77, 20020 Lainate, Italy; alessio_corazza@saes-group.com; 4Department of Medical Biotechnologies, University of Siena, 53100 Siena, Italy; gabriele.cevenini@unisi.it

**Keywords:** UV, disinfection, disinfection device, innovative techniques, UV LED, UV chip, prevention, UV technology, SARS-CoV-2, photonic measurements

## Abstract

The coronavirus SARS-CoV-2 pandemic has become a global health burden. Surface sanitation is one of the key points to reduce the risk of transmission both in healthcare and other public spaces. UVC light is already used in hospital and laboratory infection control, and some recent studies have shown its effectiveness on SARS-CoV-2. An innovative UV chip technology, described in Part I of this study, has recently appeared able to overcome the limits of old lamps and is proposed as a valid alternative to LEDs. This study was designed to test the virucidal activity on SARS-CoV-2 of a device based on the new UV chip technology. Via an initial concentration of virus suspension of 10^7.2^ TCID_50_/mL, the tests revealed a viral charge reduction of more than 99.9% after 3 min; the maximum detectable attenuation value of Log_10_ = 5.7 was measured at 10 min of UV exposure.

## 1. Introduction

The World Health Organization declared the COVID-19 pandemic on 11 March 2020. Globally, as of 12:37pm CEST, 3 April 2021, there have been 129,619,536 confirmed cases of COVID-19, including 2,827,610 deaths, reported to WHO [1]. The pandemic has continued to have a sustained trend, with the basic reproduction number (R0) estimated between 2.2 and 3.3 in the early stages of the pandemic [2,3] and a low infection fatality ratio according to the latest estimates [4].

SARS-CoV-2 is an enveloped single-stranded RNA virus of the *Coronaviridae* family and the *Nidovirales* order. Viruses in this family have similar features: a spherical shape with a diameter of 120–160 nm covered with spike proteins and with RNA of about 27–32 kb long. Some of these viruses are responsible for common colds, but MERS-CoV, SARS-CoV, and SARS-CoV-2 have a more substantial clinical impact and have caused many deaths [5,6].

Respiratory droplets are coronaviruses’ primary transmission mode but contact (either direct contact with an infected subject or indirect contact through a hand-mediated transfer of the virus from contaminated fomites to the mouth, nose, or eyes) can be another transmission route [7]. This is confirmed by evidence showing that SARS-CoV-2 can survive up to 3 h in aerosols, up to 4 h on copper, up to 24 h on cardboard, and up to 3 days on plastic and stainless steel [8], and, under some conditions, SARS-CoV-2 can survive for weeks [9,10,11]. The viability of SARS-CoV-2, as well as SARS-CoV-1, depends on the material, the environmental microclimate parameters, the medium in which the virus is deposited, and the initial viral load [12,13].

While waiting for specific drugs and an understanding of the effects of the extensive vaccination campaigns that have just begun, containment measures (social distancing, masks, and hygiene) [14,15] and tracing [16,17] are the main weapons to contain the spread of infection and will remain valid measures in every scenario, even in the future [18].

In this regard, no-touch technologies can provide mitigation practices by inactivating the virus. Their use during the COVID-19 pandemic could be one of the key points to reduce the risk of transmission both in healthcare and other public spaces. It is recognized that UVC light can stop microbial growth, and its physical approach is considered a good compromise between cost and effectiveness, which is why it is progressively becoming more and more widespread in healthcare and at home [19]. In addition, the introduction of standards such as E3131/18 (Standard Practice for Determining Antimicrobial Efficacy of Ultraviolet Germicidal Irradiation Against Microorganisms on Carriers with Simulated Soil) by The American Society for Testing and ISO 15714:2019 (Materials and the Method of Evaluating the UV dose to Airborne Microorganisms Transiting In-Duct Ultraviolet Germicidal Irradiation Devices) will standardize tests and comparisons. There are many fields of UVC applications. The main one is the disinfection of hospital or public environments and fomites [20], but the possibility of using them to disinfect and re-use PPE in case of shortages during the pandemic has also been considered [14]. Finally, the possibility of inactivating the virus in biological samples and transfusion material has been assessed [21,22].

UVC light (200–280 nm) has better germicidal properties than UVA (315–380 nm) or UVB (280–315 nm). UVC rays are absorbed by nucleic acid bases, leading to molecular structural damage through photodimerization that results in virus inactivation and an inability to replicate [14,22,23]. The effectiveness of UV on novel coronavirus is of great interest, and several studies have shown that UVC irradiation is an effective disinfection method against SARS-CoV-2 [14,19,22]. However, given the similarities described above among the various viruses of the coronavirus family, much of the evidence on the new virus is deduced from studies on surrogates and few studies have been done directly on the new coronavirus.

Different technologies produce UVC light, but most devices use low-pressure mercury lamps to produce UVC at a wavelength of 254 nm [24]. Given the known toxicity of mercury, the Minamata Convention on Mercury was signed by the United Nations Environment Program (UNEP) in 2013, and, from 2020, mercury-containing products will be banned, and new alternative UV devices should be used [25]

Among them UVC LEDs have become an alternative by solving some of the limitations of mercury lamps: a reduced size allows for a greater field of application and the absence of a warm-up time, as they are minimally affected by operating temperatures, allows for faster times of use and avoids the heating of irradiated materials [23].

Another solution can be represented by xenon pulsed light devices that generate, compared to other devices, a wider UV spectrum (200–280 nm), considered more effective by some authors [26], but they require more energy, which could lead to a reduction in lamp life.

Finally, a UV chip is cold, has a very low current, and is less influenced by temperature compared to an LED. As described in Part I of this study, it has a spatially wider (non-point-shaped) radiation source that provides a wider diffusion and energy homogeneity of light in the space. The wavelengths of the chip are centered around the maximum biocidal efficacy values of the UVC radiation (264 nm) [27]. Although there is still little evidence available, all three previous UVC technologies have proved effective in reducing the viral load of SARS-CoV-2, but our work is the first to test this chip technology on this emerging viral pathogen. This study aims to determine the virucidal activity against SARS-CoV-2 of a box with UV chips.

## 2. Materials and Methods

In June 2020, the virucidal activity of a device, based on UV chip technology by Lightlab Sweden AB, provided by SAES Getters S.p.A. was tested against SARS-CoV-2. Some of the experiments were run in the Department of Molecular and Development Medicine at the University of Siena and some were conducted in a BSL3 laboratory of the “Toscana Life Sciences Foundation” affiliated with the University of Siena.

### 2.1. Tested Device

All the UV treatments were performed in a disinfection box developed by LightLab Sweden using the newly developed UV chip technology. Six UV sources were placed in the bottom of the UVC treatment box, with each one providing 10 mW of UV power. The UV irradiation surface of the chips is a circle with a diameter of approximately 1.3 cm (a chip is about the size of a 2-euro coin). The objects to disinfect can be placed on the quartz shelf, which is positioned approximately at the mid-height of the box. A lid prevents the light from coming out. The interior of the box is carefully designed and coated with highly reflective UV paint to ensure that the UV radiation reflects and reaches, as much as possible, the objects positioned on the quartz surface. A button placed laterally on the base of the device can initiate the UV radiation of the system. Different light colors, visible around the starting button, alert the user to whether the system is in standby mode or is functioning (Figure 1a,b).

The operating principle of the UV chips differs from other commercially available sources. Briefly, electrons are accelerated in a vacuum cavity toward a material that will emit photons when struck by electrons. The spectral properties are determined by the specifics of the material. These light sources exhibit a very low working temperature. Further information or specifications on the operation principles of the UV chip are contained in Part I of this study.

### 2.2. Photometric Analysis

The photonic specifications of the box have been determined using a spectrophotometer Avantes ULS2048CL-EVO (Avantes, Apeldoorn, Netherlands) with a probe that had a cosine corrector.

The direct irradiance from the chips was evaluated at 126 points positioned at the level of the internal quartz plane (in yellow, Figure 2). To properly measure the irradiance at these points, we created a 3D-printed mask (in blue, Figure 2), using a high-resolution printer (Form 2 Formalb), which was a quarter of the size of the crystal plane. On the mask, 35 holes of the exact size of the probe with the cosine corrector were positioned to improve the precision of the measurements and to standardize the measurements and positioning. Moving the mask into the 4 different quadrants of the box, it was possible to collect the 126 measurements.

### 2.3. Setup

The UV light was activated by pressing the starting button of the white disinfection box.

Positions 1 and 2 were selected on the crystal because they do not directly face the UV chips and the UV radiation is lower in these positions; Position 1 is also a central point and where objects are usually expected to be placed (Figure 3).

These positions were inoculated with 100 µL of viral suspension of SARS-CoV-2, which had a concentration of 10^7.2^ tissue culture infective dose 50% (TCID_50_%)/mL. To better highlight the performance of the device, the suspension was placed as follows: (i) directly on the crystal, when the box was without the lid to test the direct virucide effect of the UV chips’ light coming from the base; (ii) on shielded lab slides, when the box was with the lid, to test the reflective virucide effect of the UV light reflected by the metallic layer in the box but preventing the light going through the lab slide carrier directly from the base.

### 2.4. Experimental Protocol

Each experiment was conducted in triplicate, with and without the device lid, testing 2 different positions on the quartz shelf and 3 time settings (3, 6, and 10 min). The schematic parameters are shown in Table 1.

### 2.5. Cells and Virus

All repetitions were tested for SARS-CoV-2 (Lot: VMR–SARSCPV2 VERO E6_28042020) concentration by TCID_50_% using the VERO E6 C1008 (ATCC CRL-1586) cell line.

For each experiment, the following evaluations were made: 3 samples were inoculated with the virus and subjected to the action of UV according to the protocol; 3 samples were inoculated but not treated with UV to determine the viral titer after recovery and were examined immediately after inoculation.

The collected suspensions were used to inoculate a 48-well plate into which the VERO E6 cell cultures were fixed.

Subsequent decimal dilutions were inoculated for a total of 10 dilutions. Each dilution was inoculated in 4 wells. The plates were incubated for 3 days at 37 °C ± 2 °C at 5% CO_2_ in a humidified atmosphere. After the exposure time, the residual virus activity was tested by evaluating the TCID_50_%.

### 2.6. Virus Infectivity Assays

The TCID_50_% assay was used to quantify the viral titers by determining the concentration at which 50% of the infected cells displayed a cytopathic effect (CPE).

Viral titration was determined according to the method developed by Spearman–Karber [28,29].

### 2.7. Data analysis

Excel 2016 (Microsoft Corporation, Redmond, WA, USA) was used for the data analysis and graphs. The descriptive statistics were arranged using STATA 16 SE version (StataCorp LLC, College Station, TX, USA). Preliminary data from the photometric analysis were processed with AvaSoft 8.11 (Avantes, Apeldoorn, Netherlands).

## 3. Results

### 3.1. Photometric Analysis

The measurements of the UV light spectrum of the chips are shown in Figure 4 and, as already explained in the first part of this study, it is possible to observe a relatively broad spectrum with a peak at 265 nm (UVC) and a secondary (lower) peak at approximately 300 nm (UVB) that extends to approximately 350 nm (UVA) (Figure 4).

Figure 5 shows the direct irradiance measured and modeled using the 126 sampling sites. The zones with the higher value of irradiance (max. 187.9 µW/cm^2^) were those near the corners of the box, while the lowest irradiances were measured (min. 61.9 µW/cm^2^) near to one of the long sides of the box. The center of the crystal was the part that had lower values than the corners. The light distribution was almost symmetrical with respect to the center of the system.

The average of the direct irradiance was 97.7 µW/cm^2^ with a standard deviation of 24.9 µW/cm^2^.

The average irradiance measured in the 10 holes drilled in the lid was 101.6 µW/cm^2^ with a standard deviation of 11.4 µW/cm^2^.

The average of direct irradiance in Position 1 (using 6 points) was 86.9 µW/cm^2^ with a standard deviation of 11.4 µW/cm^2^, while in Position 2 (using 6 points) it was 97.0 µW/cm^2^ with a standard deviation of 5.99 µW/cm^2^.

The average of reflected irradiance in Position 1 was 91.8 µW/cm^2^ with a standard deviation of 0.74 µW/cm^2^ and slightly higher than in Position 2 where it was 86.2 µW/cm^2^ with a standard deviation of 0.85 µW/cm^2^, unlike with direct radiation.

### 3.2. Test on SARS-CoV-2

The results for testing the device against SARS-CoV-2 were as follows:The maximum measurable Log_10_ reduction equal to 5.7 (99.9998%) was reached with an irradiation time of 10 min, for all the repetitions, regardless of the presence or not of the lid (i.e., regardless of whether the UV light was coming directly from the UV chips at the base or whether the UV light was just reflected from the reflective layers of the box);By lowering the UV exposure time to 3 min, slightly lower Log_10_ attenuation values were achieved and still greater than 3.2 (99.94%);At 6 min of exposure, the mean Log_10_ attenuation value was over 5 (99.999%);The results obtained have been schematized in Table 2 for better understanding;The dispersion graph in Figure 6 shows the surviving virus concentrations based on the radiation dose to which it was exposed.

## 4. Discussion

This device is the first UV-chip-based system, to our knowledge, to show effectiveness against SARS-CoV-2. The new UV chip technology was effective in reducing vital SARS-CoV-2 to 3.2 Logs (99.94%) after 3 min from a starting titer of 7.2 Logs requiring about 15 mJ/cm^2^, and 35 mJ/cm^2^ should be sufficient to have a reduction of TCID_50_ equal to 5.7 Log (99.9998%). Figure 6 suggests that the maximum reduction achievable with the experiment (5.7 Logs) could have been achieved with irradiation times of less than 10 min and presumably just over 6 min. These results appear to be slightly better than similar studies. For example, Fischer et al. reported a reduction of 4 Logs (99.99%) starting from 4.5 Log TCID_50_/mL in 50 uL inoculated on stainless steel with 330 mJ/cm^2^ exposure to UVC 260–285 nm [30]. According to Heilingloh et al., UVC 254 nm exposure produced a reduction of 6.7 Log TCID_50_/mL in 600 uL of viral suspension inoculated in 24 well plates with exposure to 1048 mJ/cm^2^ [19], and a deep ultraviolet light-emitting diode with 280 ± 5 nm, tested by Inagaki et al., generated an abatement of 3.2 Log PFU/mL starting from 4.3 Log PFU/mL in 150 uL spread on a 60 mm petri dish and exposed to 75 mJ/cm^2^ [22]. Hebling et al., from a review of the literature on other coronaviruses, have estimated that a dose of 3.7 mJ/cm^2^ with an upper limit of 10.6 mJ/cm^2^ is required for a 90% reduction of the coronavirus [5]. Different wavelengths may have a different efficacy on SARS-CoV-2; UVC radiation near 264 nm appears to have superior efficacy than higher wavelengths [27]. It should also be considered that the device we tested also has peaks in the UVA and UVB range that contribute to the virucidal effect albeit to a lesser degree; in fact, Heilingloh et al. reported the reduction of 1 Log at 365 nm with 292 mJ/cm^2^. However, Kitagawa et al. reported slightly better results: a 222 nm UVC light (0.1 mW/cm^2^) reduced viable SARS-CoV-2 by 2.51 Log_10_ in 30 s [31]. The effects may vary depending on the substrate considered. The radiation dose is lower in transparent substrates and thin layers, while the protein components of organic substrates, larger thicknesses, and the porosity of objects may shield the penetration of rays. In this regard, our study was carried out in a laboratory condition on a contaminated drop and not on objects taken in real public or hospital environments; however, the inoculated viral titer was very high and probably higher than what can be found in real conditions. UVC can be used to disinfect the FFP mask but a higher radiation dose is required due to the porosity of the material and the possible presence of substrates [14]. The distance of the sample from the light source can affect the experiment because the sample can be heated or can evaporate and thus the effects of UV can be overestimated; although high and prolonged temperatures are required to inactivate SARS-CoV-2 [5]. The chip tested does not heat up during operation, so the above effects, as well as damage to exposed materials caused by heat, can be considered negligible. Further studies will also be carried out to evaluate the impact of the geometry of the object on the distribution of light irradiation: larger objects can make shadows, which can decrease the reflected light, but for objects with a smaller surface the direct and reflected virucidal activity can add up and other light paths, which are difficult to identify, can increase the effect.

This device is completely new in technology and, compared to other types of devices, the chip has several strengths. Mercury technology, although still widely used, as mentioned above, will soon be replaced due to the toxicity of the metal. LEDs, pulsed-xenon UV devices, and novel chips are the main contenders. The chip, which does not even contain quartz, provides a wider diffusion and energy homogeneity of light in the space and is not affected by working temperatures. The wavelengths of the chip are centered around the maximum biocidal efficacy values of the UVC radiation (264 nm) but can cover multiple wavelengths, expanding the field of effectiveness, which is comparable to xenon technology although some lengths fall outside the UVC range. The device, which is a pre-industrial prototype, can be used in the disinfection of many objects, including medical devices of which phonendoscopes [32] can only be an example. In addition, it can also be used outside the strictly medical field in public places and even in homes.

## 5. Conclusions

In conclusion, the tests revealed a SARS-CoV-2 charge reduction of more than 99.9% after 3 min of operation. The maximum detectable attenuation of 5.7 Log (99.9998%) was measured at an irradiation time of 10 min, for all the repetitions, regardless of direct or reflected UV radiation hitting the virus samples in the device.

## Figures and Tables

**Figure 1 ijerph-18-03873-f001:**
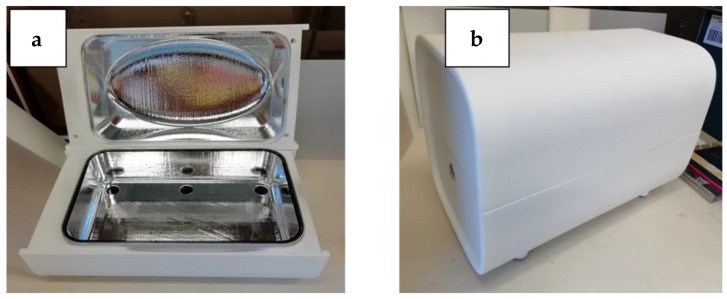
(**a**) Base and lid of the white disinfection box; UV chips are visible in the bottom part. (**b**) Box with the lid closed.

**Figure 2 ijerph-18-03873-f002:**
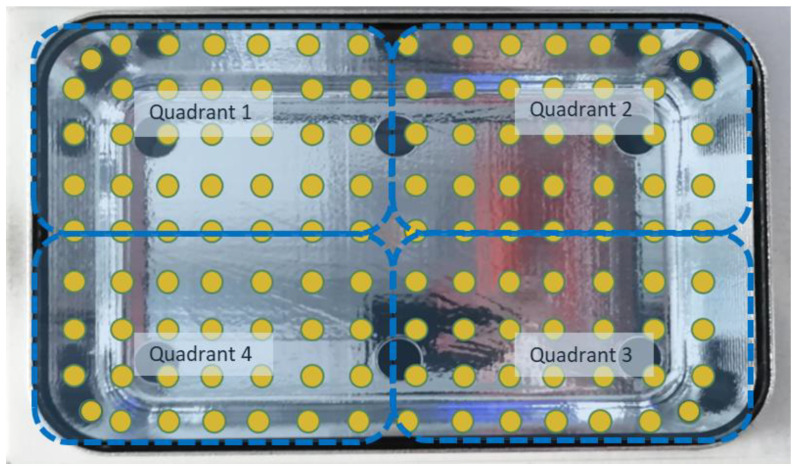
The 126 points selected on the quartz surface for photonic measurement are represented in yellow, while the dashed blue lines represent the different quadrants of the base.

**Figure 3 ijerph-18-03873-f003:**
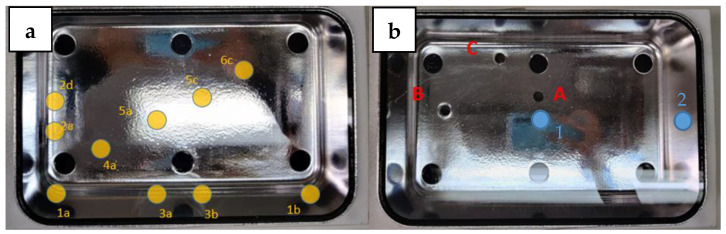
On the left (**a**), a representation of 10 points (in yellow) where the cosine corrector was inserted to measure irradiance. On the right (**b**), 3 holes (in red) drilled into the bottom of the box, and the two positions (in blue) selected to perform the tests.

**Figure 4 ijerph-18-03873-f004:**
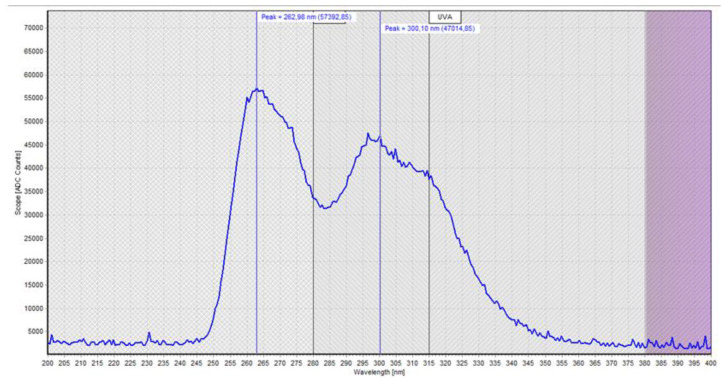
Photometric measurements of the light spectrum of the UV chip.

**Figure 5 ijerph-18-03873-f005:**
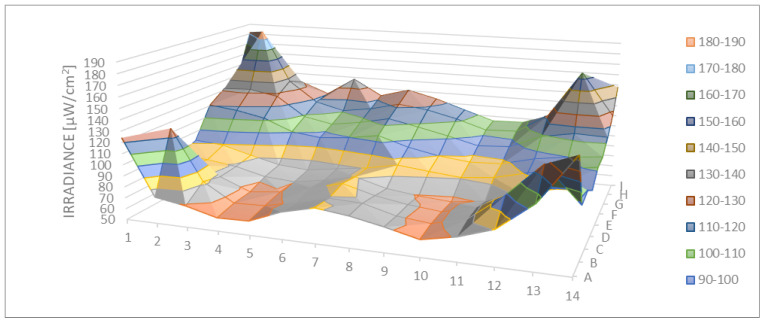
Spatial distribution of direct irradiance on the exposed surface (µW/cm^2^).

**Figure 6 ijerph-18-03873-f006:**
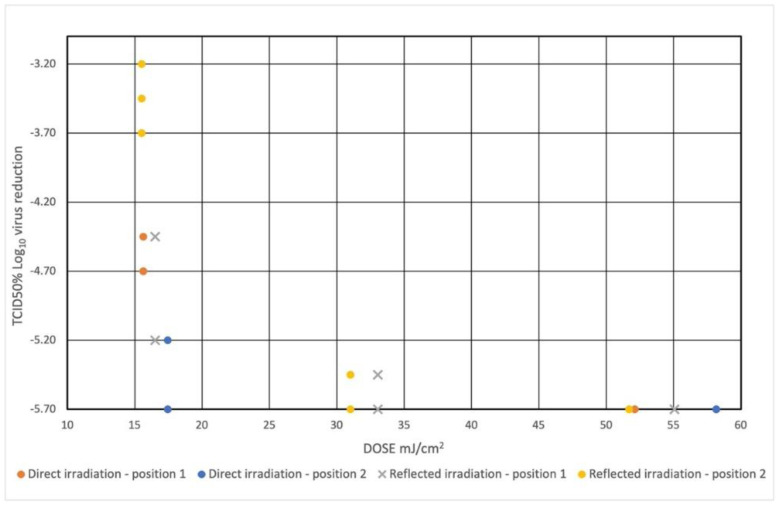
Dispersion plot shows the virus concentration reduction based on the irradiation dose to which it was exposed. The value of Log TCID_50_% = 1.50 means total viral inactivation—limit of detection. The initial concentration of viral suspension of SARS-CoV-2 was 10^7.2^ TCID_50_%.

**Table 1 ijerph-18-03873-t001:** Schematic procedure of the UV device experiment.

Device Configuration	Exposure Time	Inoculum Position
Box with lid	3 min	1; 2
	6 min	1; 2
	10 min	1; 2
Box without lid	3 min	1; 2
	10 min	1; 2

**Table 2 ijerph-18-03873-t002:** Evaluation of the virucide activity on SARS-CoV-2.

Device Configuration	Inoculum Position	Exposure Time (min)	Irradiance (µW/cm^2^)	DOSE (mJ/cm^2^)	TCID_50_% Log_10_ Mean of Untreated Virus Suspensions	TCID_50_% Log_10_ Mean of UV-Treated Virus Suspensions	TCID_50_% Log_10_ Virus Reduction Mean
With lid *	1	3	91.8	16.5	7.20	2.50	−4.70
With lid	2	3	86.2	15.5	7.20	3.75	−3.45
With lid	1	6	91.8	33.0	7.20	1.67	−5.53
With lid	2	6	86.2	31.0	7.20	1.67	−5.53
With lid	1	10	91.8	55.1	7.20	1.50	−5.70
With lid	2	10	86.2	51.7	7.20	1.50	−5.70
Without lid **	1	3	86.9	15.6	7.20	2.58	−4.62
Without lid	2	3	97.0	17.5	7.20	1.67	−5.53
Without lid	1	10	86.9	52.1	7.20	1.50	−5.70
Without lid	2	10	97.0	58.2	7.20	1.50	−5.70

* sample exposed to light reflected from the inner coating of the lid; ** sample exposed to direct chip light.

## Data Availability

The data presented in this study are available on request from the corresponding author.

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
