# Peer review of "An Emerging Innovative UV Disinfection Technology (Part II): Virucide Activity on SARS-CoV-2"

_ijerph, 2021, doi:10.3390/ijerph18083873_

Round 1

Reviewer 1 Report

An innovative and rigorously presented study. This manuscript provides a valuable scientific contribute to knowledge and management of SARS-CoV-2 in the environment and to technology advancement in the field of UV disinfection and LED applications.

Author Response

Thank you very much for your kind comment

Reviewer 2 Report

This paper reports the  an emerging innovative UVdisinfection technology (Part II): virucide activity on SARS-CoV-2. This paper can be acceptable for publication. However, I suggest that the authors should take following revisions into consideration:

Figure 6 should be improved (line contrast) , on the axis Y number format 4.5 not 4,5.

and one suggestions:

If statistical analysis was performed, the results (the significance of the differences between irradiation dose) could be intersting ?

Author Response

Reviewer 1

An innovative and rigorously presented study. This manuscript provides a valuable scientific contribute to knowledge and management of SARS-CoV-2 in the environment and to technology advancement in the field of UV disinfection and LED applications.

  • Thank you very much for your kind comment

Reviewer 2

This paper reports the  an emerging innovative UV disinfection technology (Part II): virucide activity on SARS-CoV-2. This paper can be acceptable for publication. However, I suggest that the authors should take following revisions into consideration:

Figure 6 should be improved (line contrast), on the axis Y number format 4.5 not 4,5.

  • Thank you. We have improved the image as suggested.

and one suggestions:

If statistical analysis was performed, the results (the significance of the differences between irradiation dose) could be interesting?

  • Figure 6 shows that as the dose increases, the Log TCID50% increases rapidly, in a non-linear way. On the other hand, it seems unrealistic to adapt to the data some non-linear statistical model, both for the lack of knowledge of the phenomenon (exponential?) and because there are few experimental points. Therefore, we believe that in this context, the graphical representation can describe the evidence of experimental data more simply and better than any unrealistic statistics.

Reviewer 3

I suggest to add to the introduction section information about standards (European or general) of testing disinfection methods based on UV - if applicable.

  • We recalled:

- The American Society for Testing and Materials with the E3131 18 is “Standard Practice for Determining Antimicrobial Efficacy of Ultraviolet Germicidal Irradiation Against Microorganisms on Carriers with Simulated Soil”.

- The ISO 15714:2019 (Materials and the Method of evaluating the UV dose to airborne microorganisms transiting in-duct ultraviolet germicidal irradiation devices)

In the introduction section, there is also information, that UV chip method is described in part I of this study, but I did not find the reference to this. The nearest reference (number given in the last paragraph of the introduction) is no. 27, which is the article by Gerchman Y. et al. - but it does not seem as part I of this study.

  • The citation of Gerchman on the 264 nm is correct. Part I of the paper is similarly under first revision. Since the technology has never before been published in any scientific journal, it contains its most detailed aspects (especially from the physical point of view) and describes its peculiar characteristics comparing them with those of classical UV-C LEDs, in the field of disinfection. However, the essential features of the technology useful for the full understanding of part II are described here. Nevertheless, for completeness and your comfort, we included here the abstract of Part I titled “An emerging innovative UV disinfection technology (part I): physical and microbiocidal characteristics”
  • Abstract: In this study, a novel UV disinfection technology, UV chip, is presented. Its technological, photonic and microbiocidal characteristics are evaluated taking as reference a UVC LED source of equivalent radiant power, driven at approximately 5 mW of UV output power. Microbiological inactivation levels are found to be equivalent, but the UV chip exhibits unique properties that make it applicable, where UVC LEDs present the most critical issues. Specifically, it has a radiating surface distributed on a circle of diameter 1.3 cm, a cold light emission, a near independence of performance from environmental temperature. In addition, it is portable and exhibits a broad spectrum of UV wavelengths with a peak at 264 nm where the maximum microbiocidal efficacy occurs. These properties allow its use in particular contexts where distances from the object to be disinfected can be minimized, achieving rapid and large effects. Recent developments in technology are moving towards a progressive increase in the chip’s radiant power, similar to what happens with UV LED technologies.

In the paragraph describing the experimental protocol there is information, that experiment was conducted in triplicate, in two different position and 3-time settings. Additionally, there is Table 1 with schematic procedure of the UV device experiment. I would suggest to present also the results in tabular form in the view corresponding to the protocol scheme and generally improving the way of results presentation.

  • Thank you. We have added the table with all measurements.

In the discussion I would suggest to give information if presented device is still a prototype or if it already has practical application.

  • The device is a pre-industrial prototype. We have specified this as required on line 269.

Reviewer 3 Report

I suggest to add to the introduction section information about standards (European or general) of testing disinfection methods based on UV - if applicable.

In the introduction section, there is also information, that UV chip method is described in part I of this study, but I did not find the reference to this. The nearest reference (number given in the last paragraph of the introduction) is no. 27, which is the article by Gerchman Y. et al. - but it does not seem as part I of this study.

In the paragraph describing the experimental protocol there is information, that experiment was conducted in triplicate, in two different position and 3-time settings. Additionally, there is Table 1 with schematic procedure of the UV device experiment. I would suggest to present also the results in tabular form in the view corresponding to the protocol scheme and generally improving the way of results presentation.

In the discussion I would suggest to give information if presented device is still a prototype or if it already has practical application.

Author Response

(The authors gave the same response as above.)
